# In Vitro Antibacterial Activity of Ozonated Olive Oil against Bacteria of Various Antimicrobial Resistance Profiles Isolated from Wounds of Companion Animals

**DOI:** 10.3390/ijms25063557

**Published:** 2024-03-21

**Authors:** Anna Lenart-Boroń, Klaudia Stankiewicz, Klaudia Bulanda, Natalia Czernecka, Miłosz Heliasz, Walter Hunter, Anna Ratajewicz, Karen Khachatryan, Gohar Khachatryan

**Affiliations:** 1Department of Microbiology and Biomonitoring, Faculty of Agriculture and Economics, University of Agriculture in Kraków, Adam Mickiewicz Ave. 24/28, 30-059 Kraków, Poland; klaudia.stankiewicz@student.urk.edu.pl; 2Department of Forest Ecosystems Protection, Faculty of Forestry, University of Agriculture in Kraków, 29 Listopada Ave. 46, 31-425 Kraków, Poland; 3Scientific Circle of Biotechnologists, Faculty of Biotechnology and Horticulture, University of Agriculture in Kraków, 29 Listopada Ave. 54, 31-425 Kraków, Poland; natalia.czernecka2@student.urk.edu.pl (N.C.); milosz.heliasz@student.urk.edu.pl (M.H.); waltersamuelhunter@gmail.com (W.H.);; 4Laboratory of Nanomaterials and Nanotechnology, Faculty of Food Technology, University of Agriculture, Balicka Street 122, 30-149 Kraków, Poland; karen.khachatryan@urk.edu.pl; 5Department of Food Quality Analysis and Assessment, Faculty of Food Technology, University of Agriculture, Balicka Street 122, 30-149 Kraków, Poland; gohar.khachatryan@urk.edu.pl

**Keywords:** antibacterial activity, ozone, scanning electron microscopy, topical dressings, veterinary medicine, wound infections

## Abstract

Frequent colonization and bacterial infection of skin wounds in small animals prevent or impair their healing. However, the broadly applied antimicrobial therapy of wounds is not always necessary and promotes the spread of bacterial resistance. Thus, alternatives to antimicrobial therapy, including preventive measures in the form of wound dressings with antibiotic properties, should be searched for. The aim of this study was to develop a new, efficient, cost-effective and non-toxic formulation with antimicrobial properties to serve as an alternative to antibiotic administration in wound-healing stimulation in companion animals. Nano/microencapsulated ozonated olive oil in a hyaluronan matrix was developed, with ozone concentration high enough to prevent bacterial growth. The presence and size of nano- and microcapsules were determined with scanning electron microscopy (SEM). Antibacterial activity of developed formulations was examined in vitro on 101 Gram-positive and Gram-negative bacteria isolated from the wounds of companion animals. The highest ozone concentration in the developed formulations inhibited the growth of 40.59% bacteria. Species and genus-specific differences in reactions were observed. *Enterococcus* spp. proved the least susceptible while non-pathogenic Gram-positive bacteria were the most susceptible to the examined formulations. Changes in the bacterial morphology and cell structure of *Psychrobacter sanguinis* suspension mixed with Ca-stabilized formulations with nano/microencapsulated ozonized olive oil were revealed during SEM observations. The combination of compounds that promote wound healing (hyaluronic acid, olive oil, ozone and calcium) with the antibacterial activity of the developed formula makes it a promising bionanocomposite for use as a topical dressing.

## 1. Introduction

Skin, after the gut, is the second largest organ of a vertebrate’s body and plays a major role in sensing the environment, maintaining thermal and physicochemical homeostasis and defending the organism against the environmental conditions and pathogenic microorganisms. However, any disorder of the skin function, due to, e.g., injury, leads to the formation of wounds. Importantly, skin wounds are a very common condition in small animals [1]. The wounds in animals are often colonized or infected by bacteria, affecting wound healing and increasing healthcare costs [1]. Biofilm formation by pathogenic bacteria is one of the main factors that impairs wound healing. It may cause inflammations and, in critical cases, may cause the need for the amputation of infected limbs or lead to death [2]. The prevention of biofilm formation is important in leading to the recovery of damaged skin and has a crucial therapeutic importance [3]. 

Currently, the treatment of injury and surgical site wounds in veterinary medicine involves the administration of broad-spectrum antibiotics [4]. Apart from increasing the costs of therapy and risk of adverse effects, the prolonged and unnecessary or inappropriate use of antimicrobial agents contributes to promoting antimicrobial resistance in bacteria [4]. Among the main animal pathogens that contribute to the spread of antimicrobial resistance and, at the same time, pose a threat to humans (e.g., pet owners) are *Campylobacter*, *Salmonella*, *Enterococcus*, *Staphylococcus* and *Escherichia coli* [5,6]. The majority of species typically infecting wounds of companion animals are also human pathogens [6]. The risk of antibiotic resistance transfer between animals and humans contributes to the importance of avoiding unnecessary antimicrobial treatment and searching for alternatives thereof. 

In recent years, more attention has been paid to the use of bionanomaterials in the cosmetic industry, and for medical as well as veterinary use. One of the formulations that have the potential to be used in both cosmetic and medical/veterinary applications is the combination of hyaluronic acid with ozone. The conventional use of ozone as an antibacterial agent is widely known and applied [7]. However, entrapment of ozone in olive oil allows for the formation of its derivatives that are slowly released into the intended area and, thus, have high potential as a therapeutically active ozone derivative of antibacterial properties intended for topical use [8]. In a recent study, Khachatryan et al. [2] developed a method of preparation of nano/microencapsules of ozonated olive oil in hyaluronyan matrix. The biocomposite-containing capsules proved to be stable and to exhibit slight inhibitory effect against bacteria and yeast colonizers of human skin, as well as potential pathogens. However, the observed antimicrobial effects were not strong enough for the developed formulations to be applied as a preventive measure of skin and/or wound infection. 

Therefore, in this study, we attempted to prepare the nano/microencapsulated ozonated olive oil in the concentrations that would have an antimicrobial effect strong enough to allow the developed formulations to be used as materials in the production of wound dressings intended for the prevention of wound infections in veterinary practice. 

Having the above in mind, the general goal of this study was to develop a new, efficient, cost-effective and non-toxic formulation that would act as an antimicrobial agent, thus creating an alternative to antibiotic administration in wound-healing stimulation in companion animals. In particular, the aim of the presented experiments included developing nano/microencapsulated ozonated olive oil in the hyaluronyan matrix, the concentration of which would be the most efficient in the prevention of bacterial infections in the wounds of companion animals.

## 2. Results

Previous attempts to increase the amount of ozonated oil, and, therefore, ozone, resulted in a reduction in encapsulation efficiency. However, the use of a cross-linked hyaluronic acid structure has enabled greater encapsulation of ozonated oil than was previously possible. The capsules were observed to break under electron energy and very low pressures. The successful formation of spherical nano/microcapsules containing ozonated oil at high concentrations in hyaluronic acid cross-linked with Ca^2+^ ions was confirmed by SEM images (Figure 1). Previous research has demonstrated that the concentration of added nanoemulsion has a significant impact on the size and formation of nanocapsules in hyaluronic acid [9]. The nanocapsules with the most uniform size were formed in samples with lower concentrations of propolis/oil emulsion. Conversely, higher concentrations of nanoemulsion in the sample resulted in partially formed capsules. In our previous study [2], nanocapsules of ozonated oil in hyaluronic acid were prepared and SEM microscopy revealed the presence of spherical nanocapsules containing the active substance, ranging in size from 50–100 nm, as well as single capsules measuring 150–200 nm. In this study, hyaluronic acid was cross-linked with Ca^2+^ ions to encapsulate ozonated oil at higher concentrations. This resulted in larger capsules with dimensions ranging from 700 to 1500 nm (Figure 1).

In total, 101 bacterial strains isolated from the wounds of companion animals were examined in this study. The growth of 41 (40.59%) was inhibited by the application of O2 suspension. Among 57 Gram-positive strains, the growth of 14 (24.56%) was inhibited by O1 and the growth of 20 strains was inhibited by O2. Among 44 Gram-negative strains, the growth of 14 (31.82%) and 21 (47.73%) was inhibited by O1 and O2, respectively (Table 1). 

Figure 2 shows the mean growth inhibition zones of Gram-positive and Gram-negative bacteria, caused by the application of emulsions containing O1 (smallest) and O2 (highest) concentrations. The effect of O2 is clearly stronger, whereas a comparison of the effect between Gram-positive and Gram-negative bacteria indicates slight differences between their reaction; however, these differences are not statistically significant (*p* < 0.05). In the case of O1, the mean growth inhibition of Gram-positives was 9.5 mm, whereas the one of Gram-negatives was 8.9 mm. In the case of O2, it was 13.4 and 14.04 mm for Gram-positives and Gram-negatives, respectively.

More vivid differences were observed while comparing the reaction of individual genera and species (Figure 3 and Figure 4). The least susceptible were the strains of Gram-positive *Enterococcus* spp. (no inhibition for O1 and only 0.73 mm for O2), followed by *Pseudomonas* spp. (2.4 mm for O1 and 4.6 mm for O2). As the examined bacterial strains were composed of groups of pathogenic, opportunistically pathogenic and much smaller groups of non-pathogenic Gram-positive species (such as, e.g., *Bacillus pumilus*, *Microbacterium maritypicum*, *Macrococcus luteus* or *Sporosarcina luteola*, Appendix A), their reaction to the effect of Hyal/O_3_ suspensions was compared between these groups. Interestingly, among the Gram-negative bacteria, the group of opportunistic pathogens (e.g., *Aeromonas media*, *Citrobacter freundi*, *Kocuria rhizophila*, *Psychrobacter sanguinis*) proved to be the most susceptible (6.7 mm and 8.3 mm for O1 and O2, respectively). Finally, among the Gram-positive strains, the group of non-pathogenic ones was characterized by the largest growth inhibition zones caused by the application of Hyal/O_3_ emulsions (i.e., 9.75 mm and 10.38 mm for O1 and O2, respectively).

SEM images showing a comparison of the control suspensions of *Psychrobacter sanguinis* (Figure 5A–C) with the suspension of the same bacterial strain with Ca-stabilized O_3_ formulation (Figure 5D–I) reveal evident differences in the bacterial structures, indicating effective cell wall rupture that caused increased permeability and, finally, leakage of the cell matrix.

## 3. Discussion

The concern about the spread of antibiotic resistance among pathogenic and commensal bacteria has led to increasing scientific interest in searching for preparations that would act as preventive measures against bacterial infections. In a previous study, Khaczatryan et al. [2] obtained nano- and microcapsules of ozonated olive oil in a hyaluronic acid matrix. The mild inhibitory effect of the developed formula against bacteria and *Candida*-like yeasts suggested that it can be treated as a safe ingredient in cosmetic preparations. In the current experiment, we significantly increased the concentration of ozone, which is the main bactericidal agent, in the tested formulations, starting from the concentrations of 0.33 g ozone (O1) and obtaining the final concentrations of 0.66 g (O2). Calcium ions were added to the prepared liquid formulations in order to stabilize the cross-linkage in the obtained final products. The SEM images confirmed that the nano- and microcapsules of ozonated olive oil were still present in the final, highest concentration of ozone (O2, i.e., 0.66 g). 

Hyaluronic acid is ubiquitous in vertebrates—it is involved in, e.g., cell differentiation, acts as a membrane-forming polymer, allows for tissue irrigation, provides osmotic balance, contributes to skin hydration, decreases transepidermal water loss and, finally, promotes various stages of wound healing [10]. For this reason, it has been selected as the best possible matrix in the formation of the nano/microencapsulated ozonated olive oil in our study.

Examination of the inhibitory effect of the suspensions was conducted using 101 bacterial strains, originating from the wound infections of pets (cats, dogs and rabbits) [11], including 57 Gram-positive and 44 Gram-negative bacteria. The growth inhibition of the examined bacteria varied. The O1 concentration caused growth inhibition of 14 Gram-positive and also 14 Gram-negative strains. The highest possible concentration that could be obtained still maintained the stability of formulation (O2), causing the growth inhibition of 41 (40.59%) strains. The growth inhibition zone diameters caused by O2 varied and ranged from 10 mm in two strains of *S. aureus*, *S. capitis* and *S. pseudintermedius* (Gram-positive), and *Kocuria rhizophilia* (Gram-negative) to 28 mm in *Streptococcus canis* (Gram-positive) and 25 mm in *Brevundimonas diminuta* (Gram-negative). The generally observed differences in the growth inhibition between Gram-positive and Gram-negative bacteria were statistically not significant. This was similar to the study by Serio et al. [12], who tested in vitro antibacterial activity of ozonated sunflower seed oil against Gram-negative *E. coli* and *Pseudomonas aeruginosa*, and Gram-positive *Micrococcus luteus* and *S. aureus*. However, different results were obtained by Pietrocola et al. [13], who examined the antibacterial activity of ozonized olive oil against oral and periodontal pathogenic Gram-positive and Gram-negative bacteria (*Aggregatibacter actinomycetemcomitans*, *Prevotella intermedia* and *Streptococcus mutans*). In their study, Gram-negative bacteria proved more sensitive to ozonized olive oil than Gram-positives. Such varying outcomes might be due to the fact that different bacterial species were examined in these experiments. As observed in our study, much more vivid differences in the reaction to the developed formulations were observed while comparing individual genera and/or species of bacteria. For instance, the largest growth inhibition zone diameter, which was observed for *S. canis* (Table 1), might be explained by the fact that this species has been reported as highly susceptible to treatment with both broad- and narrow-spectrum antibiotics [14,15], so the susceptibility to antimicrobial treatment may be a species-associated trait. However, in many cases, susceptibility or resistance to antimicrobial agents is associated with individual strains of bacteria. But, in this study, this could not have been verified due to the fact that there was only one *S. canis* among the tested isolates.

We divided the examined bacteria into eight groups (i.e., species of *E. coli*, genera: Gram-positive *Enterococcus*, *Staphylococcus* and *Streptococcus*; Gram-negative *Acinetobacter* and *Pseudomonas*; meanwhile, the remaining ones that were of less importance in terms of wound infection were divided into non-pathogenic Gram-positive and opportunistic pathogens of Gram-negative bacteria). Among these groups, *Enterococcus* spp. appeared the least susceptible, as none of the tested strains reacted to O1 and only one reacted to O2, thus resulting in the mean growth inhibition of 0.73 mm. Also, Acinetobacter did not react to the O1 concentration. In fact, there are very few studies demonstrating the effectiveness of ozonized oils against Enterococcus spp. and none demonstrating its effectiveness against Acinetobacter spp. This may be due to the fact that only the highest possible concentrations of ozone proved effective against these bacteria. Both ozone concentrations caused growth inhibition of Staphylococcus spp., Streptococcus spp., E. coli and Pseudomonas spp. The highest inhibition zones were observed in non-pathogenic Gram-positive bacteria and opportunistically pathogenic Gram-negative bacteria. All mentioned Gram-positive and Gram-negative genera were previously subjected to studies on their reaction to ozonated vegetable oils and showed susceptibility to the tested formulations [12,13,16]. 

The mechanism of the antibacterial activity of ozone has been widely studied and the outcomes of the studies show that its molecule releases free oxygen radicals that facilitate the formation of hydrogen peroxide and lipid peroxidation products. This results in the decreasing density and thickness of the bacterial outer membrane due to oxidative stress, resulting in the leakage of intracellular components in both Gram-positive and Gram-negative bacteria [17,18]. However, the described mechanisms of the bactericidal effect have been demonstrated in the case of gaseous ozone used in, e.g., the food industry [17,18], wastewater and drinking water treatment, medical equipment sterilization, laundry disinfection [19], or in the development of wearable gaseous ozone therapy systems [20]. An additional effect of ozone in its application in wound dressings is that the oxidative stress that it causes stimulates early wound-healing activity in cells [20]. In our study, we employed SEM microscopy to demonstrate the effect of the addition of our newly developed formulation to the suspension of bacterial cells (i.e., *Psychrobacter sanguinis*). As can be seen in Figure 5, under various magnifications, the bacterial cells, after contact with the nano/microencapsulated ozonated olive oil, become more translucent as compared to the control, suggesting the occurrence of cell deformation, cell wall damage and cell components’ leakage outside. Similar observations, but while using gaseous ozone on pathogenic bacteria such as *A. baumani*, *E. coli* and *P. aeruginosa*, were made by Rangel et al. [21], and by examining the effect of ozonated water on cell wall permeability and the ultrastructure of *P. aeruginosa* by Zhang et al. [22]. 

The formulation developed and examined in our study is a composition of four compounds, i.e., hyaluronic acid, olive oil, ozone and calcium. According to the literature data, all these components have an important role in the wound-healing process. Hyaluronic acid provides two important functions in the wound-healing process as part of cell migration and proliferation. It provides a temporary structure in the early stages of wound healing, which helps facilitate a nutrient supply and the disposal of waste products from the cell metabolism. It is involved in keratinocyte proliferation and migration. Also, because hyaluronic acid is hydroscopic, it is highly osmotic which allows for hydration during wound repair and inflammatory processes [23]. Even though the precise mechanism of olive oil involvement in wound-healing processes has not yet been elucidated, research shows that, in topical use, its phenolic compounds have anti-inflammatory effects; its polyphenols are associated with neuroprotective and antiaging effect, leading to the repair of epithelialized tissue. Topical application of olive oil has been reported to aid in angiogenesis by increasing the levels of intravascular endothelial growth factor; it can also inhibit inflammation and increase epithelial regeneration [24]. The potential therapeutic mechanism of the ozone involved in the treatment of wounds is associated with eliciting mild oxidative stress, regulating endogenous growth factors, antioxidant capacity, the modulation of hemorheology and the disinfection of the wounded area by the inactivation of pathogens [20,25]. Finally, calcium plays a vital role in extracellular signalling, is an intracellular secondary messenger modulating the proliferation, differentiation and maturation of keratinocytes and fibroblasts, and maintains epidermal lipid barrier function [26]. 

With all the above-described results, the developed formulations of nano- and microencapculated ozonated olive oil in the hyaluronan matrix proved to be a very promising bionanocomposite and an effective alternative to topical antimicrobial agents. In a previous study, Khachatryan et al. [2] demonstrated the lack of cytotoxicity in the developed formulations to human epidermal keratinocytes in the HaCat model. Further examinations of the formulations containing the highest concentration of ozone, developed in this research, would be advisable to confirm the lack of cytotoxicity and exclude any irritating effects of the formulation at the newly obtained concentration.

## 4. Materials and Methods

### 4.1. Bacterial Strains and Culture Conditions

A total of 101 bacterial strains, isolated from the wound swabs of companion animals, as described previously [11], were selected for the analysis. Briefly, the isolated bacteria were identified to the species level using MALDI-TOF (Matrix-assisted laser desorption/ionisation-time of-flight) spectrometry. Their antibiotic resistance profile was assessed using the disk-diffusion method [27], while the specific resistance mechanisms, such as ESBL (extended-spectrum beta-lactamase) in *Enterobacteriaceae* and *Pseudomonas* strains were confirmed with the double disk synergy test [28]. Resistance to macrolide, lincosamid and streptogramin B type was assessed according to [29]. The examined bacteria comprised 57 Gram-positive and 44 Gram-negative strains (Appendix A).

### 4.2. Synthesis and Characterization of Formulations Containing Nanoencapsulated Ozonated Olive Oil

High-molecular hyaluronic acid (Aquajuv CT), molecular weight 0.8–1.0 MDA; (CH_3_COO)_2_Ca (POLAURA, Zabrze, Poland); ozonated olive oil (Scandia Cosmetics S.A., Niepołomice, Poland) with an ozone content of 1.11 ± 0.02 g in 100 g of oil were used to produce the hydrogels. 

Emulsion and hydrogels were prepared using the modified method described by Khachatryan et al. [2]. The nanoemulsion was prepared by placing a mixture of 50.0 mL of water and 50.0 mL of ozonated olive oil in an ultrasonic cleaner (Polsonic, Warsaw, Poland) cooled to 5 °C and sonicated for 30 min to obtain the nanoemulsion. 

A total of 1000.0 g of a 2% solution were prepared by weighing out 20.0 g of hyaluronic acid on an analytical weight (Radwag, Białystok, Poland) and afterwards supplementing it with 970.0 mL of deionized water. Then, 10.0 g of a 5 M calcium acetate solution was added to cross-link the hyaluronic acid. The resulting suspension was stirred using a magnetic stirrer (Heidolph RZR 2020, Heidolph Instruments GmbH & Co. KG, Schwabach, Deutschland, German) until a clear gel was obtained.

Samples marked Control, O1 and O2 were obtained by adding 60.0 mL of water (Control), 30.0 mL of nanoemulsion and 30.0 mL of water (O1), 60.0 mL of nanoemulsion (O2), respectively, to 300 g of the previously obtained gel.

The mixtures were then cooled to 5 °C and homogenised using a Polytron PT 2500 E (Kinematica AG, Malters, Switzerland).

### 4.3. Antimicrobial Activity of Ozonated Olive Oil Formulations

The antimicrobial activity of the formulations containing nanoencapsulated ozonated olive oil was examined using two types of formulations. Initially, these were foils obtained as emulsions in four concentrations of ozone (0.083 g; 0.165 g, 0.33 g and 0.66 g), dried at room temperature in sterile 12 cm diameter polypropylene dishes. Due to the fact that the results of growth inhibition zones for the two lower concentrations were unsatisfactory and the growth inhibition zones obtained for foils were difficult to clearly assess, the second round of tests was conducted with liquid emulsions of the two highest concentrations (0.33 and 0.66 g of ozone).

Bacterial isolates were transferred into sterile 0.85% saline solutions to obtain 0.5 MacFarland suspensions, which were then streaked onto Mueller–Hinton agar (Argenta, Poland). Foils were cut into 5 × 5 mm squares with surface-sterilized scissors. Both squares of foils and liquid emulsions were sterilized under UV light for 30 min. 

The antibacterial activity of foils was examined by applying the 5 × 5 mm squares of foils on the surface of bacterial cultures, while the antibacterial activity of emulsions was examined by the well diffusion method (i.e., cutting wells in the agar and pouring 100 µL of suspensions into the wells).

Both types of cultures were incubated at 36 ±1 °C for 24 h. After the incubation, the results were read by measuring the growth inhibition zone diameters around the foil fragments and wells filled with emulsions. In the case of the square foils, two diameters (the smallest and the largest) were read and the final result was expressed as a mean of the two reads. All growth inhibition zones were expressed in mm.

### 4.4. Scanning Electron Microscopy (SEM) Observations

The SEM observations of nanoencapsulated ozonated olive oil interaction with bacterial cells were documented for Gram-negative *Psychrobacter sanguinis*. This species was selected based on its promising, positive reaction to the examined formulations at both concentrations (one of the largest growth inhibition zones in the in vitro test of antimicrobial activity of these formulations). Moreover, it has been reported in the literature that this bacterium can be considered an opportunistic pathogen, causing postsurgical and wound infections [30,31]. 

The samples for SEM observations were prepared by mixing 1 mL of the bacterial suspension with the density of 1 McFarland with 1 mL of Ca-stabilized nanoencapsulated ozonated olive oil liquid formulations. 

Scanning electron microscopy (SEM) was carried out with a JEOL JSM 7500F microscope (JEOL, Tokyo, Japan) coupled with an AZtecLiveLite Xplore 30 system (Oxford Instruments, Abingdon, UK). Before the analysis, the samples underwent a coating process involving a 20 nm layer of Cr using a K575X Turbo Sputter Coater (Emitech, Ashford, UK). The scanning electron microscope was equipped with a transmission electron microscope (TEM) detector. 

### 4.5. Statistical Analysis 

The experiments were carried out in triplicates. The normality of the results was examined using the Shapiro–Wilk test. As the distribution of the results was not close to normal, non-parametric tests were applied. Therefore, the Kruskal–Wallis test was used to assess the significance of differences in the following: (a) the antibacterial activity of different concentrations of ozone in the formulations, (b) antibacterial activity of formulations against Gram-positive and Gram-negative bacteria; (c) activity of formulations against bacteria belonging to different species or groups thereof. 

The significance level was set at *p* <0.05. The analyses were performed using Statistica v. 13.1 (TIBCO Software Inc., PaloAlto, Santa Clara, CA, USA).

## 5. Conclusions

In this study, we developed innovative formulations of ozonized olive oil in a hyaluronic acid matrix that contained the highest O_3_ concentrations, which remained stable. Our formula proved effective against a variety of bacteria, not only the opportunistic and mild pathogens, but also those of high potential for pathogenicity and resistance to antimicrobial agents, such as *Enterococcus* and *Acinetobacter*. The combination of four compounds, each with a proven significant role in wound-healing process, i.e., hyaluronic acid, olive oil, ozone and calcium, makes the developed formulation a promising bionanocomposite for its future use as an alternative to topical antimicrobial agents. Their application as, e.g., wound dressings would enhance tissue regeneration and prevent the wound area from being colonised by pathogenic microorganisms.

## Figures and Tables

**Figure 1 ijms-25-03557-f001:**
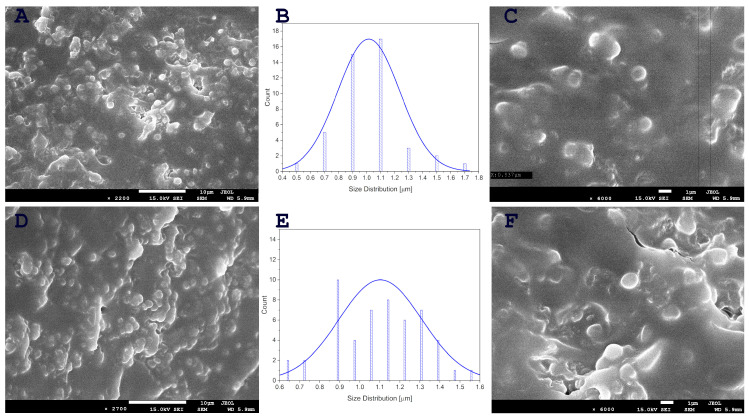
SEM images and capsule-size distribution of samples O1 (**A**–**C**) and O2 (**D**–**F**).

**Figure 2 ijms-25-03557-f002:**
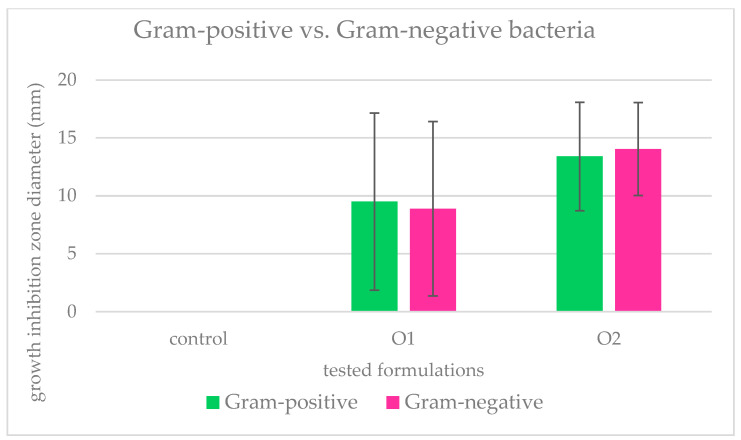
Mean growth inhibition zones (bars represent standard deviations) for Gram-positive and Gram-negative bacteria caused by the application of suspensions with two concentrations of Hyal/O_3_ nano/microcapsules. The results are means of three replications (*p* < 0.05).

**Figure 3 ijms-25-03557-f003:**
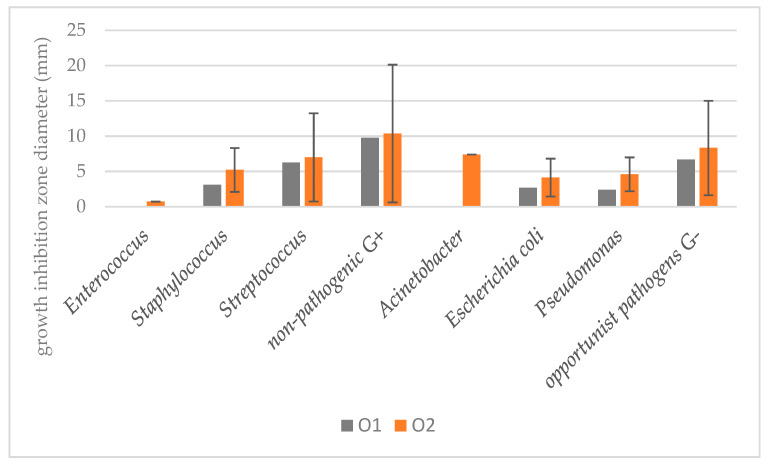
Mean growth inhibition zones (bars represent standard deviations) for the selected bacterial species and genera of pathogenic and non-pathogenic Gram-positive and Gram-negative bacteria. The results are means of three replications (*p* < 0.05).

**Figure 4 ijms-25-03557-f004:**
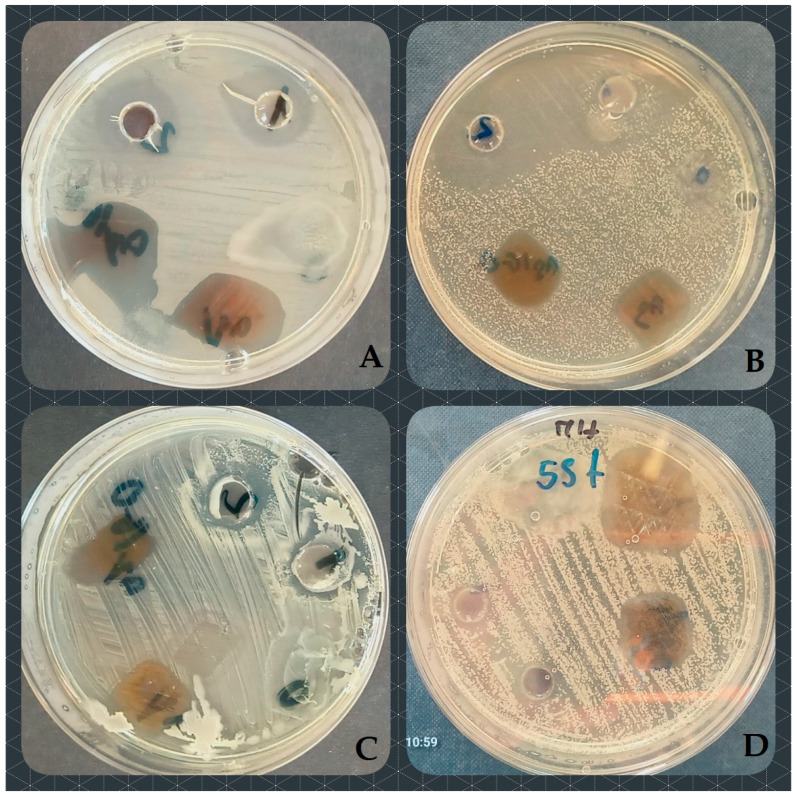
Growth inhibition caused by the suspensions containing Hyal/O_3_ nano/microcapsules against (**A**) *E. coli* (growth inhibition zones: 11 mm [O1] and 12 mm [O2]); (**B**) *Streptococcus canis* (growth inhibition zones: 25 mm [O1] and 24 mm [O2]); (**C**) *P. sanguinis* (growth inhibition zones: 17 mm [O1] and 17 mm [O2]); and (**D**) *S. lentus* (growth inhibition zones: 12 mm [O1] and 13 mm [O2]).

**Figure 5 ijms-25-03557-f005:**
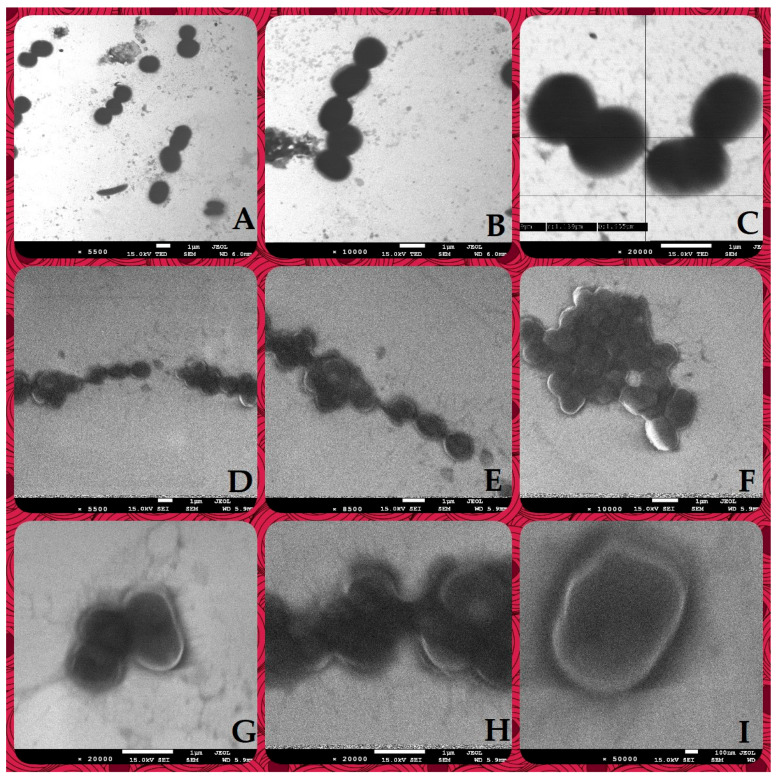
SEM images of mixtures of *P. sanguinis* with control suspensions (**A**–**C**) and Ca-stabilized nano and microcapsules of Hyal/O_3_. (**D**–**I**). The magnifications are as follows: 5500 (**A**), 10,000 (**B**), 20,000 (**C**), 5500 (**D**), 8500 (**E**), 10,000 (**F**), 20,000 (**G**), 20,000 (**H**), 50,000 (**I**).

**Table 1 ijms-25-03557-t001:** Growth inhibition zones for the bacterial strains as a result of nano/microcapsules’ application in two concentrations.

Species	O1	O2	Control
Gram-positive (*n* = 20)			
*Enterococcus faecalis*	0	11	0
*Lysinibacillus fusiformis*	11	12	0
*Macrococcus canis*	12	12	0
*Microbacterium oxydans*	11	13	0
*Micrococcus luteus*	10	11	0
*Peribacillus simplex*	12	13	0
*Sporosarcina luteola*	22	22	0
*Staphylococcus aureus*	8	10	0
*Staphylococcus aureus*	0	10	0
*Staphylococcus capitis*	0	10	0
*Staphylococcus devriesei*	15	16	0
*Staphylococcus felis*	10	11	0
*Staphylococcus lentus*	12	13	0
*Staphylococcus pseudintermedius*	0	11	0
*Staphylococcus pseudintermedius*	0	11	0
*Staphylococcus pseudintermedius*	0	10	0
*Staphylococcus pseudintermedius*	11	12	0
*Staphylococcus pasteuri*	20	20	0
*Staphylococcus sciuri*	11	11	0
*Streptococcus canis*	25	28	0
Gram-negative (*n* = 21)			
*Acinetobacter calcoaceticus*	0	12	0
*Acinetobacter ursingii*	0	12	0
*Acinetobacter ursingii*	0	13	0
*Brevundimonas diminuta*	20	25	0
*Enterobacter hormaechei*	11	15	0
*Enterobacter hormaechei*	20	20	0
*Escherichia coli*	8	12	0
*Escherichia coli*	0	11	0
*Escherichia coli*	10	11	0
*Escherichia coli*	11	12	0
*Kocuria rhizophilia*	0	10	0
*Moraxella osloensis*	13	16	0
*Proteus mirabilis*	12	12	0
*Proteus mirabilis*	15	20	0
*Pseudomonas plecoglossicida*	0	11	0
*Pseudomonas putida*	12	12	0
*Psychrobacter pulmonis*	15	23	0
*Psychrobacter sanguinis*	0	12	0
*Psychrobacter sanguinis*	17	17	0
*Serratia liquefaciens*	0	11	0
*Serratia marcescens*	14	15	0

## Data Availability

The data presented in this study are available on request from the corresponding author.

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
