# Peer review of "In Vitro Antibacterial Activity of Ozonated Olive Oil against Bacteria of Various Antimicrobial Resistance Profiles Isolated from Wounds of Companion Animals"

_ijms, 2024, doi:10.3390/ijms25063557_

Round 1

Reviewer 1 Report

Comments and Suggestions for Authors

Comments:

This study tackles the significant challenge of skin wounds in small animals, which are commonly affected by bacterial colonization and infection, leading to impaired wound healing and fostering bacterial resistance to antibiotics. The aim was to propose an alternative to antibiotic therapy by formulating nano-ozonated olive oil encapsulated within a hyaluronic matrix with elevated ozone concentrations. The overall combination of wound-healing agents and antibacterial properties within the developed formulation suggests its potential as a promising topical treatment for managing wounds in small animals.

Despite the promising objectives of this study and the thorough preparation undertaken, there are certain observations and recommendations that should be incorporated to enhance the quality of this research. Hence, we believe that a major revision of the manuscript is necessary before reaching a final decision.

Abstract:

1.     Remove the sentence "Skin wounds are very common in small animals." since the statement applies broadly, indicating that wounds are prevalent not only among small animals.

2.     Rewrite the background of the abstract more effectively.

3.     Rephrase the aims of the study in the abstract.

Materials and Methods:

4.     In subsection 4.3, it is mentioned by the author(s) that "flakes are acquired from emulsions at four different concentrations," yet specific details regarding these concentrations are not provided.

5.     In subsection 4.4., The author(s) indicated that "Observations of the interaction between nanoencapsulated ozonated olive oil and bacterial cells were conducted using two selected bacterial species: Gram-positive Staphylococcus lentus and Gram-negative Psychrobacter sanguinis." What was the rationale behind choosing these specific bacterial species for observing the interaction with nanoencapsulated ozonated olive oil instead of other species?

Results:

6.     What evidence from the SIM test demonstrated that the micro- and nano-spherical capsules possess the highest concentration of ozone and are cross-linked with Ca2+ ions?

7.     The author(s) did not provide information regarding the size of the micro- and nano-spherical capsules observed through SEM.

8.     Include the dimensions of the micro- and nano-spherical capsules in Figure 1.

9.     In line 94, the author(s) stated that the quantity of tested Gram-positive bacteria was 57, whereas in Table 1, the sample size (n) was indicated as 20.

10.  Display the statistical function or symbol denoting statistical significance on Figures 1 and 2.

11.  In table 1, What is the reason for Streptococcus canis exhibiting the largest growth inhibition zone compared to other bacteria?

12.  Figure 4, illustrating Zone Inhibition, lacks clarity as it is unclear which shape corresponds to "E. coli," "Streptococcus spp.," "P. sanguinis," or "S. lentus.", Furthermore, there were no specifics provided regarding the bacterial type or the dimensions of the zones in Figure 4.

13.  In the materials and methods section, it was stated that both Gram-positive Staphylococcus lentus and Gram-negative Psychrobacter sanguinis were utilized to observe the interaction between nanoencapsulated ozonated olive oil and bacteria. However, Figure 5 only displayed the combinations of P. sanguinis with control suspensions.

The reviewer

Author Response

Dear Reviewer,

thank You for this review, we revised the manuscript based on your comments. The answered are mentioned below.

Comments and Suggestions for Authors

Comments:

This study tackles the significant challenge of skin wounds in small animals, which are commonly affected by bacterial colonization and infection, leading to impaired wound healing and fostering bacterial resistance to antibiotics. The aim was to propose an alternative to antibiotic therapy by formulating nano-ozonated olive oil encapsulated within a hyaluronic matrix with elevated ozone concentrations. The overall combination of wound-healing agents and antibacterial properties within the developed formulation suggests its potential as a promising topical treatment for managing wounds in small animals.

Despite the promising objectives of this study and the thorough preparation undertaken, there are certain observations and recommendations that should be incorporated to enhance the quality of this research. Hence, we believe that a major revision of the manuscript is necessary before reaching a final decision.

Abstract:

  1. Remove the sentence "Skin wounds are very common in small animals." since the statement applies broadly, indicating that wounds are prevalent not only among small animals.

Reply: The sentence was removed and the fragment rephrased

  1. Rewrite the background of the abstract more effectively.

Reply: the background was rewritten

  1. Rephrase the aims of the study in the abstract.

Reply: The aims were rephrased.

Materials and Methods:

  1. In subsection 4.3, it is mentioned by the author(s) that "flakes are acquired from emulsions at four different concentrations," yet specific details regarding these concentrations are not provided.

Reply: The concentrations of ozone were provided in brackets and the fragment was rephrased.

  1. In subsection 4.4., The author(s) indicated that "Observations of the interaction between nanoencapsulated ozonated olive oil and bacterial cells were conducted using two selected bacterial species: Gram-positive Staphylococcus lentus and Gram-negative Psychrobacter sanguinis." What was the rationale behind choosing these specific bacterial species for observing the interaction with nanoencapsulated ozonated olive oil instead of other species?

Reply: The species were selected based on their reaction to the application of ozone formulations, i.e. they were characterized by one of the largest growth inhibition zones in an in vitro test. However, we also changed this fragment, due to the fact that SEM images were satisfactory (high quality pictures with clearly marked differences between control and ozone application) only for P. sanguinis, therefore we changed information in the Materials and Methods section that the interactions between bacteria and ozonated oil formulations were documented for Gram-negative P. sanguinis. Also this bacterium has been reported to cause postsurgical and wound infections, so it perfectly corresponds to the scope of this research. We added an explanation and citations.

Results:

  1. What evidence from the SIM test demonstrated that the micro- and nano-spherical capsules possess the highest concentration of ozone and are cross-linked with Ca2+ ions?

Reply: New SEM images have been added to the work. A section of the text has been rephrased. The experiments conducted enabled us to encapsulate a greater amount of ozonated oil than we had previously achieved. However, the capsules obtained are not very stable under drastic conditions such as high vacuum and electron bombardment, making it rare to obtain images of whole capsules. The capsules break under the influence of electron energy and very low pressures, as seen in previous images.

  1. The author(s) did not provide information regarding the size of the micro- and nano-spherical capsules observed through SEM.

Reply: The distribution of capsule sizes was provided, along with new SEM images.

  1. Include the dimensions of the micro- and nano-spherical capsules in Figure 1.

Reply: The section containing the drawing and text has been rewritten.

  1. In line 94, the author(s) stated that the quantity of tested Gram-positive bacteria was 57, whereas in Table 1, the sample size (n) was indicated as 20.

Reply: the total number of Gram-positive bacteria tested in this study was 57 and their entire list is provided in the Supplementary Table 1. Table 1 in the text provides a total of 20 Gram-positive strains, the growth of which was inhibited by the O2 formulation.

  1. Display the statistical function or symbol denoting statistical significance on Figures 1 and 2.

Reply: The size and diameter distribution of the capsules are presented in Figure 1, 2 and 3.

  1. In table 1, What is the reason for Streptococcus canis exhibiting the largest growth inhibition zone compared to other bacteria?

Reply: the high susceptibility can be both species-specific as well as strain-specific. We found reports in the literature that this species is highly susceptible to antibiotic treatment, so the susceptibility to the action of antimicrobial agents can of course be species-specific. On the other hand, the susceptibility or resistance to antimicrobial agents can vary within individual species and be strain-specific. However, this could not have been verified within this study, as the species that we examined were isolated from wounds of animals, and this particular species has been isolated only once throughout the study. We added an explanation to this in the text.

  1. Figure 4, illustrating Zone Inhibition, lacks clarity as it is unclear which shape corresponds to "E. coli," "Streptococcus spp.," "P. sanguinis," or "S. lentus.", Furthermore, there were no specifics provided regarding the bacterial type or the dimensions of the zones in Figure 4.

Reply: the letters A, B, C and D were added to the Figure to show the corresponding bacterial strains. The growth inhibition zone diameters were provided in the figure caption, too.

  1. In the materials and methods section, it was stated that both Gram-positive Staphylococcus lentus and Gram-negative Psychrobacter sanguinis were utilized to observe the interaction between nanoencapsulated ozonated olive oil and bacteria. However, Figure 5 only displayed the combinations of P. sanguinis with control suspensions.

Reply: The fragment of Materials and Methods was rephrased. We obtained satisfactory (high quality) pictures only for P. sanguinis, therefore we selected only these to document the interaction between our formulations and bacteria. Also this bacterium has been reported to cause postsurgical and wound infections, so it perfectly corresponds to the scope of this research. We added an explanation and citations.

Thank You again for Your valuable comments, thanks to which the work will probably be more readable and valuable.

With kind regards,

Authors

Reviewer 2 Report

Comments and Suggestions for Authors

Comments and questions

1. A supplementary explanation is needed for the results of SEM images in Figure 1. It is difficult to clearly understand what Figure 1 is trying to explain.

2. In Figure 2 and Figure 3, the value of standard deviations for the Y-axis (growth inhibition zone diameter) is significantly larger compared to the main values. Is it appropriate? (in particular, for O1 in Figure 1 and for Figure 3)

-------------------------------------------------The end---------------------------------------

Comments on the Quality of English Language

Overall, I think it is well written. Please check again for typos or awkward phases.

Author Response

Comments and questions

  1. A supplementary explanation is needed for the results of SEM images in Figure 1. It is difficult to clearly understand what Figure 1 is trying to explain.

Previous attempts to increase the amount of ozonated oil, and therefore ozone, resulted in a reduction in encapsulation efficiency. However, the use of a cross-linked hyaluronic acid structure has enabled greater encapsulation of ozonated oil than was previously possible. The capsules were observed to break under electron energy and very low pressures, as seen in previous SEM images. New microscopic images were taken, revealing spherical structures. Obtaining stable images of whole capsules is difficult due to their instability under drastic conditions such as high vacuum and electron bombardment. Whole capsule images are rarely obtained as a result.

  1. In Figure 2 and Figure 3, the value of standard deviations for the Y-axis (growth inhibition zone diameter) is significantly larger compared to the main values. Is it appropriate? (in particular, for O1 in Figure 1 and for Figure 3)

This is exactly what happened. The standard deviations were very large, particularly for the O1 concentration, because the growth inhibition zones varied largely. As can be observed in Table 1, the results can vary significantly, and for the O1 concentration, the growth inhibition ranged from none (0 mm) to 25 mm. For this reason, we concluded that the O2 concentration is more reliable and effective in inhibiting the growth of bacteria.

-------------------------------------------------The end---------------------------------------

Comments on the Quality of English Language

Overall, I think it is well written. Please check again for typos or awkward phases.

Response: done

With kind regards,

Authors

Reviewer 3 Report

Comments and Suggestions for Authors

Briefly about the main problems:
- The novelty remained unclear to me.
- There are no control experiments.
- Organic oil is not capable of storing ozone, oil is very well oxidized by ozone
- Very poor arsenal of methods (only two)
- Looks more like a promotional item rather than a scientific article

Details:
Novelty. The authors take hyaluronic acid, add a divalent cation and obtain micro- and nano-objects. Obviously, everything is as it should be and this has been known for a long time! The fact that the authors add a drop of oil is essentially not new. Maybe the authors have invented some new approach? Also no!

Control experiments The authors say that hyaluronic acid, calcium cations and oil have a significant antibacterial effect. What exactly has the effect? Control experiments are needed separately with hyaluronic acid, separately with calcium cations and separately with oil. Then with combinations of reagents...

About ozone.
Ozone is not stored in oil. Ozone, even in the refrigerator, will react with oil within an hour. Moreover, he will react completely, without a trace. The authors use commercial ozonated oil. This means that days pass between the ozonation process and use. The authors simply use seriously oxidized oil. In this case, a control experiment is needed with undamaged oil... This also explains the absence of any measurements of ozone concentration.

Very poor arsenal of methods.
The entire manuscript is based on electron microscopy and microbiological tests. Obviously, this is very little. + Some procedures are described in very general terms...

- Looks more like a promotional item rather than a scientific article Let me give you a specific example. The authors write in the first sentence of the results: “The SEM documentation (Figure 1) confirmed the successful formation of spherical micro- and nanocapsules containing ozone at the highest concentration, cross-linked with Ca2+ ions.” How can you confirm that these are capsules using SEM? That they contain high concentrations of ozone? And that all this is cross-linked with calcium? By the way, I didn’t find nanoobjects in the photographs... Then comes the second sentence...

Round 2

Reviewer 1 Report

Comments and Suggestions for Authors

No additional comments

Reviewer 2 Report

Comments and Suggestions for Authors

The authors' responses to the reviewer's questions were thought to be clear and conscientious. And it seems to have been edited to reflect those contents well in the text. Therefore, this manuscript is considered acceptable in this journal.

Reviewer 3 Report

Comments and Suggestions for Authors

Novelty. The authors take hyaluronic acid, add a divalent cation and obtain microobjects. Obviously, everything is as it should be and this has been known for a long time! The fact that the authors add a drop of oil is essentially not new. Maybe the authors have invented some new approach? Also no!

Response: Previous attempts to increase the amount of ozonated oil, and therefore ozone, resulted in a reduction in encapsulation efficiency. However, the use of a cross-linked hyaluronic acid structure has enabled greater encapsulation of ozonated oil than was previously possible. The capsules were observed to break under electron energy and very low pressures, as seen in previous SEM images. New microscopic images were taken, revealing spherical structures. Obtaining stable images of whole capsules is difficult due to their instability under drastic conditions such as high vacuum and electron bombardment.

Commentary on the authors' response.
The authors' response strengthened my opinion that the novelty of the result was dubious. The authors added hyaluronic acid, which is known, and obtained “greater encapsulation of ozonated oil than was previously possible.” I believe that research is based on novelty of the third kind...

Control experiments The authors say that hyaluronic acid, calcium cations and oil have a significant antibacterial effect. What exactly has the effect? Control experiments are needed separately with hyaluronic acid, separately with calcium cations and separately with oil. Then with combinations of reagents...

Response: also, as explained before, the control has been provided to every step of the experiment. As stated in Materials and methods (4.2), nanoemulsion with ozone were the examined substances, while gel without the nanoemulsion was treated as control. All controls have also been used in the previous study (https://doi.org/10.3390/ijms232214005), which describes the development of the emulsion. The emulsion described in this study is characterized by an increased content of ozone, as also described in the manuscript text. Calcium ions were used to stabilize (cross-link) the suspensions. Then, control was used in culture-based assessments and in SEM imaging. In the developed formulation, the three ingredients (hyaluronic acid, calcium and olive oil) are
characterized by wound healing promotion, not bactericidal properties. For this reason, their addition in the formulation is not aimed at increasing the antibacterial effect, unlike ozone.

Commentary on the authors' response.
The authors should conduct additional control experiments. And don’t write about how all the experiments have supposedly been carried out. It's not good to mislead the reviewer!

About ozone. Ozone is not stored in oil. Ozone, even in the refrigerator, will react with oil within an hour. Moreover, he will react completely, without a trace. The authors use commercial ozonated oil. This means that days pass between the ozonation process and use. The authors simply use seriously oxidized oil. In this case, a control experiment is needed with undamaged oil... This also explains the absence of any measurements of ozone concentration.

Response: as stated earlier:
Ozonated oils have been recognised for a considerable time. Depending on their botanical origin and the amount of unsaturated bonds, they can bind varying amounts of ozone in the form of ozonides, which exhibit sufficient stability [1-3]. In this study, ozonated olive oil was used and the ozone content was determined [4] according to the European Pharmacopoeia (European Pharmacopoeia, 2019.(Ph. Eur.) 10th edition | EDQM - European Directorate for
the quality of medicines) before emulsion preparation and encapsulation. The ozone content was also determined after six months of storage, and the decrease was found to be less than 1%. Some authors [5,6] have reported that ozonised oil can remain stable for even longer periods, up to 8 years.

Commentary on the authors' response.
The authors are again trying to deceive the reviewer and readers. There is no ozone in oil and there cannot be!!!! There may be ozonides, this has nothing to do with ozone. Even ozonides of normal structure are not very stable. The authors can easily check this, just like half a century ago, ozonides have characteristic bands in the IR spectra in the region of 980-110 cm-1. Please do additional experiments if you insist on your point of view.

Very poor arsenal of methods. The entire manuscript is based on electron microscopy and microbiological tests. Obviously, this is very little. + Some procedures are described in very general terms...

We have previously studied the physicochemical properties of the resulting capsules, the encapsulation process, and the properties of the gels and composites (including other polysaccharides). In the previous (cited) study (https://doi.org/10.3390/ijms25063121), bacteria that were subjected to the tests have been isolated, identified based on phenotypic and proteomic (MALDI-TOF) methods, followed by the antibiotic resistance tests and
molecular (PCR) detection of genetic determinants of antibiotic resistance. This has also been stated in section 4.1. In this work, our focus is on investigating the effect and mechanism of action of the developed materials on the previously described bacteria. We also expanded the description of the conducted analyses. Also, the description of synthesis and characterization of the developed formulation is very precise because this is the very essence
of the novelty described in this manuscript, i.e. the development of the formulation, the antibacterial properties of which were examined. The antibacterial testing procedures or SEM microscopy are golden standards of laboratory work, the details of which do not have to be described in details, only the relevant references are needed.

Commentary on the authors' response.
There is no need to respond to this comment; the experimental arsenal was and remains poor!

- Looks more like a promotional item rather than a scientific article Let me give you a specific example. The authors write in the first sentence of the results: “The SEM documentation (Figure 1) confirmed the successful formation of spherical micro- and nanocapsules containing ozone at the highest concentration, cross-linked with Ca2+ ions.” How can you confirm that these are capsules using SEM? That they contain high concentrations of ozone? And that all this is cross-linked with calcium? By the way, I didn’t find nanoobjects in the photographs... Then comes the second sentence...

The cross-linking of hyaluronic acid with various ions, particularly divalent ions like Ca2+ and Mg2+, has been a well-known phenomenon for a long time [Giubertoni et al., 2021]. However, the novelty lies in the use of a cross-linked form of hyaluronic acid to encapsulate more ozonated oil. Giubertoni, G., Pérez de Alba Ortíz, A., Bano, F., Zhang, X., Linhardt, R. J., Green, D. E., ... & Bakker, H. J. (2021). Strong reduction of the chain rigidity of hyaluronan by selective binding of Ca2+ ions. Macromolecules, 54(3), 1137-1146. New SEM images have been added to the work. A section of the text has been rephrased.
The experiments conducted enabled us to encapsulate a greater amount of ozonated oil than we had previously achieved. However, the capsules obtained are not very stable under drastic conditions such as high vacuum and electron bombardment, making it rare to obtain images of whole capsules. The capsules break under the influence of electron energy and very low pressures, as seen in previous images. Nanostructures refer to the thin layers, typically only a few nanometres thick, that are visible in SEM images of fractured capsules

Commentary on the authors' response.
Nanoobjects either did not exist or did not exist. It's good that they rephrased the sentence...